# ENTROPY-AWARE SPECULATIVE DECODING TOWARD IMPROVED LLM REASONING

## ABSTRACT

Speculative decoding (SD) accelerates large language model (LLM) reasoning by using a small draft model to generate candidate tokens, which the target LLM either accepts directly or regenerates upon rejection. However, excessive alignment between the draft and target models constrains SD to the performance of the target LLM. To address this limitation, we propose Entropy-Aware Speculative Decoding (EASD), a training-free enhancement. Building on standard SD, EASD incorporates a dynamic entropy-based penalty. At each decoding step, we employ the entropy of the sampling distribution to quantify model uncertainty. When both models exhibit high entropy with substantial overlap among their top-N predictions, the corresponding token is rejected and re-sampled by the target LLM. This penalty prevents low-confidence errors from propagating. By incorporating draft-model verification, EASD enables the possibility of surpassing the target model's inherent performance. Experiments across multiple reasoning benchmarks demonstrate that EASD consistently outperforms existing SD methods and, in most cases, surpasses the target LLM itself. We further prove that the efficiency of EASD is comparable to that of SD. The code can be found in the Supplementary Materials.

## 1 INTRODUCTION

Large language models (LLMs), such as GPT-4 (Achiam et al., 2023), Qwen (Bai et al., 2023), and LLaMA (Touvron et al., 2023), have demonstrated remarkable performance across diverse natural language processing (NLP) tasks. However, their reasoning relies on autoregressive decoding, where each token is generated through a full forward pass conditioned on all previous tokens. This strict sequential dependency leads to high latency in large-scale models and long-text generation, making computational cost a central bottleneck for practical deployment (Patterson et al., 2021; Frantar et al., 2022; Lin et al., 2024). To mitigate this issue, speculative decoding (SD) (Chen et al., 2023a) has emerged as an effective paradigm for reducing reasoning cost while preserving quality. Its core idea is to use a small draft model to propose multiple candidate tokens, which the larger target model then verifies or corrects in parallel within a single forward pass. This substantially reduces the number of target model invocations and overall latency. Building on standard SD, methods such as Fast Inference (Leviathan et al., 2023), Medusa (Cai et al., 2024), Hydra (Ankner et al., 2024), and EAGLE (Li et al., 2024a;b; 2025b) further introduce mechanisms like multi-token verification, parallel decoding heads, and uncertainty-guided token selection, continuously advancing SD's acceleration potential. Nonetheless, output quality is ultimately capped by the target model, because the final predictions must match its judgments.

To improve the quality of SD, recent studies have proposed integrating reward models into the decoding process (Li et al., 2025a; Liao et al., 2025). These reward models are trained to approximate human preferences or task-specific correctness (Wei et al., 2023) and evaluate sequences or individual steps during reasoning. However, reward models are neural networks that require additional forward passes, substantially increasing the computational cost of SD (Gao et al., 2023; Lambert et al., 2024). Moreover, most operate at the sequence or step level, making fine-grained token-level control and collaborate difficult. In the broader area of efficient LLM reasoning, researchers have explored dynamic allocation of computation based on input difficulty or token importance. For example, Mixture-of-Experts (MoE) models (Lepikhin et al., 2020) selectively activate a subset of expert layers per token, reducing computational cost while preserving model capacity. Early-

exit methods (Liu et al., 2024; Chen et al., 2023b) allow tokens to stop processing once sufficient confidence is reached, and routing LLMs (Ong et al., 2024; Jitkrittum et al., 2025) learn policies to assign inputs to the most appropriate submodels. While these architectures enable scalable and efficient computation, they often require substantial structural modifications and are difficult to integrate into existing decoding pipelines. Furthermore, their reliance on training with new data can limit generalization to broader tasks or domains.

To address these issues, we propose a training-free enhancement to SD called Entropy-Aware Speculative Decoding (EASD). As shown in Figure 1, EASD builds on the standard token acceptance and rejection strategy. It introduces a dynamic, entropy-driven penalty that is applied only to tokens meeting specific conditions. At each decoding step, we measure model uncertainty using the entropy of the sampling distribution. If both the draft and target models exhibit high entropy and their Top-N candidate tokens substantially overlap, the token is rejected and re-sampled by the target model. This mechanism prevents low-confidence tokens from propagating errors during reasoning. As a result, it reduces the risk of incorrect final outputs. By leveraging draft-model verification, EASD can potentially surpass the inherent performance of the target model. It provides a lightweight, efficient, and practical way to improve SD.

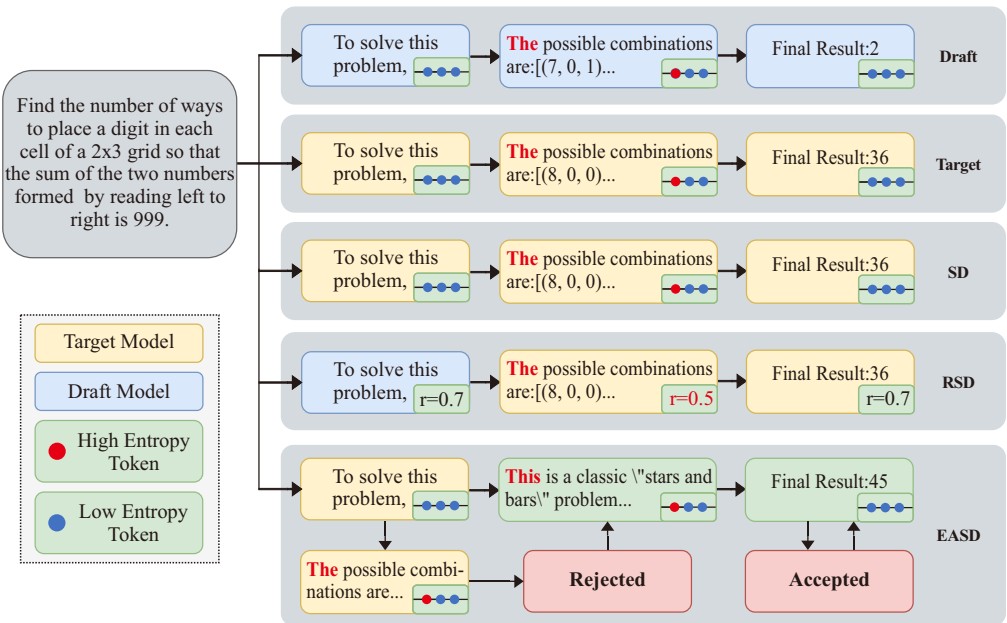

Figure 1: Comparison of traditional SD, the state-of-the-art method RSD, and our proposed EASD. Traditional SD enforces strict alignment with the target model, while RSD selectively accepts outputs from the draft model. However, when the target model itself produces suboptimal tokens, neither SD nor RSD can correct or redirect the output. EASD overcomes this limitation by introducing token-level adjustments. For example, when both the draft and target models exhibit high entropy on the token *The*, EASD rejects it and chooses *This* instead. This change enables a new generation path, *This is a classic 'stars and bars' problem*, instead of directly enumerating the answer as in the original output.

We evaluate EASD on several challenging reasoning benchmarks, including Olympia (He et al., 2024), MATH500 (Hendrycks et al., 2021), and GPQA-Diamond (Rein et al., 2024). These tasks require high token-level accuracy, where even small deviations can lead to incorrect answers. Experimental results show that EASD significantly improves generation accuracy while retaining the inference speedups of standard SD. Compared to heuristic baselines and reward-model-based filtering methods, EASD achieves higher output performance with lower computational cost. Moreover, EASD outperforms the target large model on multiple benchmarks, demonstrating its potential to surpass the model's inherent performance.

To summarize, our main contributions are:

- We propose a training-free enhancement to SD that introduces a dynamic, entropy-guided penalty on target model tokens, facilitating more effective collaboration between draft and target models.
- By leveraging draft-model verification, EASD prevents error propagation from low-confidence tokens, significantly improving token-level accuracy while preserving the reasoning speedups of standard SD.
- Experiments on multiple challenging reasoning benchmarks show that EASD not only outperforms existing speculative decoding methods but also surpasses the inherent performance ceiling of the target LLM.

## 2 RELATED WORK

This paper reviews related research on enhancing LLM reasoning in three major areas: collaboration between large and small language models, speculative decoding, and entropy-based inference.

### 2.1 COLLABORATION BETWEEN LARGE AND SMALL LANGUAGE MODELS

In recent years, collaborative frameworks between large and small language models have garnered increasing attention. Their goal is to balance computational efficiency with generation quality. Existing approaches mainly rely on coarse-grained interactions. In these methods, small models act as lightweight assistants to preprocess inputs. For example, they may reformulate user queries into more informative prompts or generate intermediate annotations to reduce the computational burden on larger models (Chen et al., 2025; Shen et al., 2024; Yang et al., 2025; Pan et al., 2025).

However, these approaches generally treat language models as independent modules connected sequentially via prompt-based information transfer. They rely on each model's individual contextual capabilities. This limits deep collaboration at the decoding level. As a result, the potential for joint optimization is constrained.

To overcome this limitation, recent studies have focused on finer-grained collaboration mechanisms. A common strategy is to interleave or fuse token distributions among models during decoding. This allows multiple models to jointly participate in token-level generation (Hao et al., 2025; Bian et al., 2025). These approaches offer opportunities for seamless integration and dynamic division of labor.

Nevertheless, efficiently implementing token-level fusion remains challenging. It is still difficult to overcome the inherent performance bottlenecks of large and small models. In this work, we further explore token-level collaborative decoding between large and small models. Our goal is to develop a more flexible and efficient approach to token generation.

### 2.2 SPECULATIVE DECODING

Speculative decoding has become a key technique for accelerating inference through collaboration between large and small language models. It maintains generation quality while reducing computation (Chen et al., 2023a). In this framework, a lightweight draft model generates candidate tokens. A more powerful target model then verifies or rejects these candidates to ensure consistency with the target distribution. This design has inspired various optimizations. These include improved candidate generation, adaptive verification thresholds, and hybrid schemes (Sun et al., 2024; Ankner et al., 2024; Cai et al., 2024; Li et al., 2025b;a). The goal is to reduce calls to the target model while preserving output quality.

However, most existing approaches assume that the draft model must be highly aligned with the target model. Some recent studies challenge this assumption. They consider partially aligned or deliberately misaligned draft models. This allows emphasis on controllability, stylistic preferences, or task-specific accuracy (Bachmann et al., 2025; Liao et al., 2025). In these cases, the target model mainly ensures linguistic fluency and global coherence. This can enable the system to outperform the target model alone. These findings suggest that speculative decoding is not only an acceleration tool. It is also a broader framework for improving generation quality through asymmetric collaboration.

A key limitation of prior work is its reliance on trained evaluation models. These models make token-level decisions but are highly data-sensitive. This limits generalization across datasets or dif-

ferent combinations of large and small models. To address this, we introduce an entropy-based strategy. It enables efficient collaboration without training. We do not require strict alignment between the small and large models. Instead, we amplify tokens in which the small model is confident. Tokens with low confidence from the large model are constrained. This allows small and large models to complement each other at the token level. It leverages their strengths and mitigates low-quality outputs. The result is a simple and effective improvement in reasoning quality.

## 2.3 ENTROPY-BASED INFERENCE

Entropy has long been recognized as a fundamental signal of model uncertainty (Wang et al., 2025). A growing body of work demonstrates its usefulness in guiding inference strategies across various generation tasks (Volkenstein, 2009). Early studies applied entropy-based measures to adjust sampling temperatures, regulate beam search heuristics, or set adaptive thresholds for candidate acceptance. These approaches improved the balance between diversity and precision (Simonds, 2025).

More recent methods extend this idea to dynamic model routing. Here, token-level entropy determines whether a lightweight model suffices or if a stronger model is needed. Entropy is also used in ensemble decoding, where token distributions from multiple sources are reweighted based on their uncertainty (Zhang et al., 2024; Qiu et al., 2024). These studies consistently show that entropy is a powerful indicator for balancing efficiency, reliability, and exploration in language generation.

Despite some successes, entropy-driven mechanisms have not yet been systematically integrated into speculative decoding frameworks. Incorporating tailored entropy-based uncertainty estimation into the draft–verify pipeline can enable more adaptive interactions. Specifically, we invoke verification only for high-uncertainty tokens and dynamically reallocate tasks between the draft and target models. This approach combines uncertainty-aware inference with the efficiency advantages of speculative decoding. As a result, it improves both efficiency and generation quality compared to existing entropy-guided strategies.

## 3 ENTROPY-AWARE SPECULATIVE DECODING

In this section, we present **Entropy-Aware Speculative Decoding (EASD)**, a novel, training-free extension of standard speculative decoding. EASD enhances the reliability and quality of generated outputs by leveraging entropy-based signals from both the draft and target models. It dynamically adjusts the target model's token selection under high uncertainty and distributional similarity, using a targeted penalty mechanism. This approach mitigates error propagation from the draft model, improving decoding efficiency and output coherence without additional training.

### 3.1 FORMULATION

The key idea of speculative decoding is to generate a draft sequence speculatively and then accept as many prefix tokens as possible based on a verification step. This method leverages the fact that draft models can often predict correctly for several steps, especially when they approximate the target model well.

Let $p_t(\cdot \mid \mathbf{x})$ denote the conditional probability distribution of the target model given a prefix $\mathbf{x}$, and $p_d(\cdot \mid \mathbf{x})$ denote that of the draft model. The draft model generates a speculative sequence of candidates $c_1, c_2, \ldots, c_k$, where each $c_i$ is sampled as: $c_i \sim p_d(\cdot \mid \mathbf{x}, c_1, \ldots, c_{i-1})$. Here, $i$ is the speculation length, typically chosen based on model sizes and desired speedup.

During the verification phase, the target model computes probabilities for all drafted tokens in parallel. Each candidate $c_i$ is either accepted or rejected based on the agreement between the two models. Formally, token $c_i$ is accepted only if all previous candidates are also accepted and

$$\epsilon_i \leq \frac{p_t(c_i \mid \mathbf{x}, c_1, \ldots, c_{i-1})}{p_d(c_i \mid \mathbf{x}, c_1, \ldots, c_{i-1})}, \qquad \epsilon_i \sim \mathcal{U}(0, 1). \tag{1}$$

Specifically, $\epsilon_i$ is sampled uniformly from $(0, 1)$. If the probability of token $c_i$ under the target model exceeds that under the draft model, the token is accepted. Otherwise, it is accepted with a probability that decreases as the gap between the two probabilities increases. If the token is rejected,

the procedure resorts to direct sampling from the target model, drawing a correction token from $p_t(\cdot \mid \mathbf{x}, c_1, \ldots, c_{i-1})$.

## 3.2 OVERVIEW

Speculative decoding accelerates LLM inference by using a smaller draft model to propose candidate tokens, which the larger target model verifies through a single forward pass. At step $t$, the draft model samples a token from its distribution $P_d^{(t)}$ over $\mathcal{V}$, and the target model accepts or rejects the token based on agreement. This reduces computation but risks potential errors. The risk is amplified when both the draft and target models are uncertain, as they may align in their uncertainty and lead the verification step to accept suboptimal tokens.

EASD improves this by integrating entropy-based uncertainty and distributional overlap. It monitors entropy in both models and overlap in their top-$n$ tokens. When uncertainty is high and overlap is significant, EASD penalizes the target model's probability for the draft's top token, promoting alternative selections. This enhances robustness without altering model training.

## 3.3 ENTROPY COMPUTATION

To measure uncertainty at step $t$, we use Shannon entropy for the draft model's distribution $P_d^{(t)}$:

$$H_d^{(t)} = -\sum_{i \in \mathcal{V}} P_d^{(t)}(i) \log P_d^{(t)}(i), \tag{2}$$

and similarly for the target model:

$$H_t^{(t)} = -\sum_{i \in \mathcal{V}} P_t^{(t)}(i) \log P_t^{(t)}(i). \tag{3}$$

High entropy reflects greater uncertainty, corresponding to a flatter distribution, whereas low entropy indicates higher confidence with a more peaked distribution.

## 3.4 TRIGGERING CONDITIONS FOR DYNAMIC PENALTY

EASD applies a penalty only when two specific conditions are met: high entropy and top-$n$ overlap. This combination identifies scenarios where the draft model is likely to mislead the target model. The rationale is that when models from the same family exhibit high uncertainty yet similar token distributions, it often indicates ambiguous decision points where errors are prone to propagate.

1. **High-Entropy Condition:** Both models show high uncertainty if:

$$H_d^{(t)} > \tau_H \quad \text{and} \quad H_t^{(t)} > \tau_H, \tag{4}$$

    where $\tau_H$ is a tunable threshold. High entropy indicates a flat probability distribution, meaning the model lacks a clear preference, which makes its decision susceptible to minor perturbations.

2. **Top-$n$ Overlap Condition:** The overlap ratio between top-$n$ token sets $T_d^n$ (draft) and $T_t^n$ (target) is:

$$\text{Overlap}(T_d^n, T_t^n) = \frac{|T_d^n \cap T_t^n|}{n}, \tag{5}$$

    (*e.g.*, $n = 5$). The condition triggers if $\text{Overlap} > \tau_O$, where $\tau_O \in [0, 1]$. This assesses distributional similarity, highlighting potential alignment issues.

## 3.5 APPLICATION OF THE DYNAMIC PENALTY

If both conditions are met, EASD rejects the draft's proposed token $t_d$ by setting:

$$P_t^{(t)}(t_d) = 0, \tag{6}$$

and renormalizing:

$$P_t^{(t)}(i) = \frac{P_t^{(t)}(i)}{\sum_{j \neq t_d} P_t^{(t)}(j)}, \quad \forall i \neq t_d. \tag{7}$$

This forces the target model to sample alternatives, reducing error risks and improving coherence.

## 3.6 THRESHOLD SELECTION

Proper selection of the thresholds $\tau_H$ (entropy) and $\tau_O$ (top-$n$ overlap) is critical for the performance of EASD, as they directly determine when the dynamic penalty is applied. We adopt a systematic, data-driven procedure to set these thresholds without introducing additional training.

**Entropy Threshold $\tau_H$.** For each validation sample $s \in \mathcal{D}_{\text{val}}$ and decoding step $t \in \{1, \ldots, T_s\}$, the target model produces a probability distribution $P_{s,t}(\cdot)$ over the vocabulary $\mathcal{V}$. The Shannon entropy is defined as

$$H_{s,t} = -\sum_{v \in \mathcal{V}} P_{s,t}(v) \log P_{s,t}(v). \tag{8}$$

We collect all entropy values across the validation set into

$$\mathcal{E} = \{H_{s,t} : s \in \mathcal{D}_{\text{val}}, \ t = 1, \ldots, T_s\}. \tag{9}$$

Let $p \in (0, 1)$ denote the proportion of high-entropy steps considered ($p = 0.05$). We identify the subset $\mathcal{S}_p \subseteq \mathcal{E}$ corresponding to the top $p$ fraction of entropy values, *i.e.*, $|\mathcal{S}_p| = \lceil p \cdot |\mathcal{E}| \rceil$. The entropy threshold is then defined as the mean entropy of these uncertain steps:

$$\tau_H = \frac{1}{|\mathcal{S}_p|} \sum_{H_{s,t} \in \mathcal{S}_p} H_{s,t}. \tag{10}$$

This construction ensures that $\tau_H$ reflects a representative level of uncertainty by focusing on the most ambiguous decoding steps, while maintaining sufficient statistical robustness.

**Top-$n$ Overlap Threshold $\tau_O$.** Complementary to the entropy-based criterion, the overlap threshold $\tau_O$ regulates cases where the draft and target models produce highly similar candidate distributions. We fix $\tau_O = 0.8$, meaning that when at least four out of the top-5 tokens coincide between the two models, the dynamic penalty is triggered. This setting reflects the intuition that excessive alignment between the draft and target models reduces the diversity and corrective potential of speculative decoding, and thus requires additional penalization to prevent redundant acceptance.

## 4 EXPERIMENTS

### 4.1 EXPERIMENTS SETTINGS

**LLMs.** To assess the performance of EASD, we employ Qwen2.5-7B-Instruct, Qwen2.5-32B-Instruct, and Qwen2.5-72B-Instruct models as our draft and target models. For experiments involving PRM, we select the Skywork-o1-Open-PRM-Qwen-2.5-1.5B model.

**Datasets.** We evaluate our method on a diverse set of reasoning tasks, including OlympiadBench (He et al., 2024), MATH500 (Hendrycks et al., 2021), AIME24 (MAA, 2024), AMC23 (MAA, 2023), GPQA-Diamond (Rein et al., 2024), and Minerva Math (Lewkowycz et al., 2022).

**Baselines.** We categorize the baselines into four groups: **(1) Target model only.** This baseline executes the target model independently, which typically results in higher computational cost compared to EASD. **(2) Draft model.** This category consists of common test-time scaling techniques built upon the draft model, including draft model only, direct generation, majority voting, and beam search. For majority voting and beam search, we employ a large number of samples, even exceeding the cost of using the target model alone, to approximate their converged performance. **(3) Speculative Decoding (SD).** We further consider speculative decoding, a method proposed for accelerating inference (Chen et al., 2023a). **(4) Reward-Guided Speculative Decoding (RSD).** RSD integrates a process reward model to score intermediate decoding steps and adaptively determine whether the target model should be invoked, thus striking a balance between efficiency and output quality (Liao et al., 2025).

**Implement Details.** All experiments are conducted on 8 NVIDIA A800 GPUs. For majority voting, beam search, and direct generation, we set the temperature to 0.7 and top$-p$ to 0.8, while the remaining methods are configured with temperature = 0 and top$-p$ = 1. Specifically, for RSD, we define a generation terminated by $\backslash n \backslash n$ as a reasoning step, which is then evaluated by a PRM with a threshold of 0.7. For EASD, the entropy threshold is determined based on the LIMO (Ye et al., 2025) dataset, where we calculate the average entropy of the top 5% highest-entropy tokens and use this value as the cutoff.

Table 1: Accuracy results on benchmark datasets, where TM denotes the target model, DM denotes the draft model, and PRM denotes the process reward model.

| Method | TM | DM | PRM | Olympiad | Minerva | Math500 | AMC23 | Aime24 | GPQA | Avg |
|---|---|---|---|---|---|---|---|---|---|---|
| Single Model | 32B | - | - | 46.81 | 44.49 | 82.2 | 60 | 16.67 | 45.96 | 49.35 |
| Single Model | - | 7B | - | 35.70 | 37.13 | 73.2 | 50 | 10 | 36.36 | 40.39 |
| Majority Voting (N=16) | - | 7B | - | 45.33 | 43.01 | 81.0 | 67.5 | 16.67 | 41.41 | 49.15 |
| Beam Search (N=16) | - | 7B | - | 40.15 | 39.34 | 78 | 62.5 | 10 | 42.42 | 45.40 |
| SD | 32B | 7B | - | 45.33 | 45.22 | 80.8 | 57.5 | 13.33 | 49.49 | 48.61 |
| RSD (Liao et al., 2025) | 32B | 7B | 1.5B | 46.5 | 46.32 | **83.8** | **67.5** | 16.67 | 50.50 | 51.88 |
| EASD | 32B | 7B | - | **48.15** | **47.06** | 80.8 | 62.5 | **23.33** | **55.55** | **52.89** |

| Method | TM | DM | PRM | Olympiad | Minerva | Math500 | AMC23 | Aime24 | GPQA | Avg |
|---|---|---|---|---|---|---|---|---|---|---|
| Single Model | 72B | - | - | 44.59 | 44.85 | **84** | 65 | 16.67 | 46.96 | 50.33 |
| Single Model | - | 7B | - | 35.70 | 37.13 | 73.2 | 50 | 10 | 36.36 | 40.39 |
| Majority Voting (N=16) | - | 7B | - | 45.33 | 43.01 | 81.0 | 67.5 | 16.67 | 41.41 | 49.15 |
| Beam Search (N=16) | - | 7B | - | 40.15 | 39.34 | 78 | 62.5 | 10 | 42.42 | 45.40 |
| SD | 72B | 7B | - | 44.15 | 44.12 | 83.8 | 60 | 13.33 | 50.50 | 49.31 |
| RSD (Liao et al., 2025) | 72B | 7B | 1.5B | **45.62** | **46.32** | 83.8 | 65 | 16.67 | 47.48 | 50.81 |
| EASD | 72B | 7B | - | 44.74 | 44.85 | 83.6 | **67.5** | **20** | **52.02** | **52.12** |

## 4.2 RESULTS ANALYSIS

The experimental results in Tables 1 provide a comprehensive comparison between our proposed **EASD** and several baselines under two different model scales (Qwen2.5-32B/72B as the target model and Qwen2.5-7B as the draft model). We observe three key findings.

First, **EASD consistently achieves the best overall performance**. In the 32B setting, EASD reaches an average score of **52.89**, which is +3.54 higher than the strongest single 32B model (49.35) and +3.74 higher than majority voting (49.15). Similarly, in the 72B setting, EASD attains an average score of **52.12**, outperforming the single 72B model (50.33) and all other baselines. This indicates that our entropy-aware rejection strategy effectively leverages the cooperation between large and small models, yielding gains that cannot be achieved by naive ensemble or decoding strategies.

Second, when compared with the recent **RSD** (Liao et al., 2025) approach, which relies on an additional reward model to partially guide alignment, EASD exhibits *both higher accuracy and greater stability*. Specifically, EASD improves over RSD by +1.01 (32B setting) and +1.31 (72B setting) in average performance. While RSD shows improvements over standard speculative decoding (SD), its dependence on a reward model introduces alignment noise and computational overhead, which limits its effectiveness. In contrast, EASD achieves superior results without requiring any reward model, showing that carefully constraining rejection strategies at the token level is a reliable and lightweight alternative.

Third, **task-level analysis** highlights the robustness of EASD. On the challenging *AIME24* benchmark, EASD significantly outperforms all baselines, achieving **23.3** (32B) and **20.0** (72B), compared to only 16.67 for single models and RSD. Similarly, on *GPQA-Diamond*, which evaluates fine-grained knowledge reasoning, EASD reaches **55.55** (32B) and **52.02** (72B), substantially higher than all alternatives. Notably, EASD maintains strong performance on *Math500* (80.8/83.6) and *AMC23* (62.5/67.5), matching or exceeding the best baselines, which suggests that our method generalizes well across both symbolic reasoning and knowledge-intensive tasks.

Overall, these results demonstrate that EASD offers a lightweight yet effective improvement to speculative decoding. By reducing dependence on reward models while maintaining strong accuracy across diverse reasoning benchmarks, EASD provides a practical step forward in multi-model cooperation.

## 4.3 ABLATIOIN EXPERIMENTS

To better understand the contribution of each component in our proposed method, we conduct ablation studies with the following variants: **(1) EASD**: The complete version of our proposed Entropy-Aware Speculative Decoding. **(2) EASD (Without Overlap)**: A variant that removes the overlap-based regulation, in which the penalty no longer depends on the token-level similarity between the

draft model and the target model. **(3) EASD (Without DM Entropy)**: A variant that excludes the entropy signal from the draft model, so the adjustment is guided only by the target model. **(4) EASD (Without DM Entropy + Without Overlap)**: A more simplified variant where both the draft model entropy signal and the overlap-based regulation are removed. These ablations allow us to examine the relative importance of draft model entropy and overlap regulation in achieving the final performance improvements.

The results in Table 2 demonstrate the effectiveness of jointly leveraging both the draft model entropy and the overlap-based regulation. The complete version of EASD consistently achieves the best overall performance across benchmarks for both the 32B and 72B target models. When overlap regulation is removed, performance declines moderately, suggesting that overlap contributes to stabilizing token selection but is less critical than entropy. In contrast, removing the draft model entropy $H_d$ leads to the most significant degradation, especially in average accuracy, indicating that $H_d$ provides essential uncertainty signals that guide effective speculative decoding. Furthermore, when both $H_d$ and overlap regulation are excluded, the performance drops further, confirming that these two components play complementary roles. Overall, the ablation study highlights that integrating both entropy signals and overlap is crucial for maximizing the benefits of speculative decoding.

Table 2: Ablation Study on the Effects of Draft Model Entropy and Overlap Regulation.

| Method | TM | DM | Olympiad | Minerva | Math500 | AMC23 | Aime24 | GPQA | Avg |
|---|---|---|---|---|---|---|---|---|---|
| EASD | 32B | 7B | **48.15** | **47.06** | **80.8** | 62.5 | **23.33** | **55.55** | **52.89** |
| w/o Overlap | 32B | 7B | 47.41 | 44.85 | 80.8 | **65** | 20 | 52.53 | 51.77 |
| w/o $H_d$ | 32B | 7B | 46.52 | 44.85 | 80.2 | 62.5 | 20 | 51.01 | 50.85 |
| w/o ($H_d$ + Overlap) | 32B | 7B | 47.41 | 46.32 | 80.6 | 60 | 13.33 | 48.48 | 49.56 |

| Method | TM | DM | Olympiad | Minerva | Math500 | AMC23 | Aime24 | GPQA | Avg |
|---|---|---|---|---|---|---|---|---|---|
| EASD | 72B | 7B | **44.74** | **44.85** | **83.6** | 67.5 | **20** | 52.02 | **52.12** |
| w/o Overlap | 72B | 7B | 44.59 | 44.49 | 82.8 | **70** | 16.67 | **53.03** | 51.93 |
| w/o $H_d$ | 72B | 7B | 43.85 | 44.12 | 83.0 | 67.5 | 16.67 | 50 | 50.86 |
| w/o ($H_d$ + Overlap) | 72B | 7B | 44.15 | 44.49 | 82.4 | 62.5 | 16.67 | 46.97 | 49.53 |

## 4.4 COMPUTATION ANALYSIS

Tables 3 report the average number of generated tokens across benchmark datasets, which serves as a proxy for computational cost. We highlight three main observations.

First, its efficiency is on par with standard speculative decoding (SD): token usage remains almost identical in both 32B and 72B settings, indicating that entropy-aware rejection incurs negligible overhead. Second, it outperforms reward-based speculative decoding (RSD) in efficiency. The reliance on a process reward model consistently increases FLOP computation for RSD, whereas EASD avoids this cost while retaining accuracy gains. Finally, EASD exhibits stable token consumption across benchmarks. Even on datasets requiring longer generations, its overhead does not exceed that of SD or RSD, and in certain cases it is slightly lower.

Table 3: Average tokens on benchmark datasets, where TM denotes the target model, DM denotes the draft model, and PRM denotes the process reward model.

| Method | TM | DM | PRM | Olympiad | Minerva | Math500 | AMC23 | Aime24 | GPQA | Avg |
|---|---|---|---|---|---|---|---|---|---|---|
| Single Model | 32B | - | - | 775.33 | 573.11 | 565.07 | 736.68 | 828.7 | 532.26 | 668.525 |
| Single Model | - | 7B | - | 842.55 | 661.44 | 606.08 | 838.55 | 977.23 | 589.07 | 752.49 |
| SD | 32B | 7B | - | 776.43 | 585.15 | 565.57 | 773.25 | 823.27 | 524.64 | 674.72 |
| RSD (Liao et al., 2025) | 32B | 7B | 1.5B | 808.49 | 628.86 | 582.17 | 752.0 | 866.43 | 489.97 | 687.97 |
| EASD | 32B | 7B | - | 775.74 | 567.70 | 554.59 | 799.70 | 867.97 | 502.72 | 678.07 |

| Method | TM | DM | PRM | Olympiad | Minerva | Math500 | AMC23 | Aime24 | GPQA | Avg |
|---|---|---|---|---|---|---|---|---|---|---|
| Single Model | 72B | - | - | 941.30 | 667.25 | 628.38 | 877.23 | 1256.4 | 772.28 | 857.14 |
| Single Model | - | 7B | - | 842.55 | 661.44 | 606.08 | 838.55 | 977.23 | 589.07 | 752.49 |
| SD | 72B | 7B | - | 943.23 | 669.99 | 632.52 | 852.5 | 1342.77 | 752.75 | 866.46 |
| RSD (Liao et al., 2025) | 72B | 7B | 1.5B | 885.0 | 655.50 | 637.41 | 835.25 | 1061.27 | 732.44 | 801.15 |
| EASD | 72B | 7B | - | 911.64 | 654.18 | 641.13 | 899.85 | 1287.73 | 748.37 | 857.15 |

In summary, EASD achieves accuracy improvements without additional computational burden, confirming its effectiveness as a lightweight alternative to reward-model-based approaches.

## 4.5 EMPIRICAL ANALYSIS OF HIGH-ENTROPY TOKEN DISTRIBUTION

To conduct the empirical analysis, we examine the probability distribution of high-entropy tokens before and after applying the EASD method, highlighting their variations and potential implications. As shown in Table 4, the original distribution exhibits relatively moderate spreads: *"Given"* ranks the highest at 2.34%, followed by *"will"* and *"Let"* (both 1.55%), *"need"* (1.53%), and *"Thus"* (1.50%). Other tokens, such as *"calculate"* (1.11%), *"Now"* (1.08%), *"consider"* (1.01%), and *"determine"* (0.87%), appear with gradually decreasing probabilities, reflecting a balanced distribution centered on initial conditions (*"Given"*), assumptions (*"Let"*), and basic reasoning steps (*"First"*, *"Thus"*).

After applying EASD, the distribution undergoes distinct adjustments. Notably, *"Given"* increases from 2.34% to 2.64%, reinforcing its position as the most salient high-entropy token and suggesting stronger emphasis on premise-related information. Similarly, *"Thus"* rises from 1.50% to 2.27%, indicating enhanced focus on conclusion-drawing, while *"consider"* grows from 1.01% to 1.77%, reflecting a shift toward analytical reasoning.

In contrast, some tokens decline: *"will"* drops from 1.55% to 1.00% and *"Let"* decreases from 1.55% to 1.33%, suggesting reduced reliance on speculative or hypothetical phrasing. Furthermore, originally low-probability tokens such as *"need"*, *"Now"*, and *"determine"* are removed from the high-entropy set. They are replaced by new entries like *"understand"* (0.97%), *"Using"* (0.92%), and *"use"* (0.91%), which highlight comprehension and method application.

Overall, EASD effectively reallocates the probability mass of high-entropy tokens: it strengthens focus on logical premises and deductive conclusions while introducing tokens related to analytical and practical processes. This modulation indicates that EASD not only stabilizes reasoning coherence but also guides the model toward more operationally useful problem-solving behaviors.

Table 4: Probability distribution of high-entropy tokens before and after applying EASD.

| Token | Original (%) | EASD (%) | Change |
|---|---|---|---|
| Given | 2.34 | 2.64 | ↑ Stronger focus on premises |
| Thus | 1.50 | 2.27 | ↑ Stronger focus on conclusions |
| consider | 1.01 | 1.77 | ↑ More analytical reasoning |
| Let | 1.55 | 1.33 | ↓ Less hypothetical usage |
| will | 1.55 | 1.00 | ↓ Less speculative focus |
| need | 1.53 | – | Removed from high-entropy set |
| Now | 1.08 | – | Removed from high-entropy set |
| determine | 0.87 | – | Removed from high-entropy set |
| understand | – | 0.97 | New emphasis on comprehension |
| Using | – | 0.92 | New emphasis on method application |
| use | – | 0.91 | New emphasis on method application |

## 5 CONCLUSION AND LIMITATIONS

In this work, we propose Entropy-Aware Speculative Decoding (EASD), a training-free extension of speculative decoding that leverages entropy signals from both draft and target models to prevent low-confidence tokens from propagating errors. EASD retains the efficiency advantages of standard SD while achieving substantial improvements in token-level accuracy across challenging reasoning benchmarks. Our results demonstrate that EASD not only outperforms existing speculative decoding variants but also has the potential to surpass the inherent performance of the target model. These findings highlight EASD as a lightweight, effective, and broadly applicable enhancement to efficient LLM inference.

Although EASD shows consistent improvements over recent speculative decoding, our study has several limitations. First, the evaluation has not been extended to broader real-world reasoning scenarios, which may involve more diverse tasks and constraints. Second, our experiments primarily focus on medium- to large-scale LLMs, leaving the effectiveness of EASD on smaller, larger, and domain-specific LLMs for future work. Finally, while EASD is training-free, its interaction with advanced alignment techniques and domain-adapted fine-tuning remains underexplored.

ETHICS STATEMENT

Our study focuses on developing and evaluating a decoding strategy for LLMs using publicly available reasoning benchmarks. No human subjects or sensitive personal data were involved. The research does not involve applications that could cause harm, discrimination, or privacy violations. We used LLMs solely for model inference and algorithmic experimentation. All experiments were conducted responsibly, and the results are reported transparently to avoid potential misuse of the evaluated models.

REPRODUCTION STATEMENT

To facilitate reproducibility, all datasets, benchmarks, and evaluation metrics used in this study are described in detail in the main text and Supplementary Materials. Our implementation of EASD, as well as baseline Speculative Decoding methods, is provided in the Supplementary Materials. For each experiment, we specify model checkpoints, hyperparameters, and decoding settings, allowing other researchers to replicate our results and extend our approach to other LLMs and reasoning tasks.

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

## A  USE OF LLMS

Large language models (LLMs) are employed in this study solely to aid and polish the writing process. Specifically, LLMs assist in refining the clarity, coherence, and grammatical accuracy of the manuscript, while all substantive content, experiments and analyses are developed by the authors.

## B  PSEUDOCODE FOR ENTROPY-AWARE SPECULATIVE DECODING (EASD)

The following pseudocode summarizes the workflow of **Entropy-Aware Speculative Decoding (EASD)**, an extension of standard speculative decoding (SD). EASD retains the core two-model structure of SD, where a draft model proposes candidate tokens and a target model verifies them. The key innovation of EASD is the *dynamic penalty* mechanism, which is applied when both models exhibit high entropy and their top-$n$ token predictions significantly overlap. This mechanism suppresses the draft model's top token and renormalizes the target distribution, encouraging alternative selections and reducing error propagation. Red text in the pseudocode highlights these differences relative to conventional SD.

---

**Algorithm 1** Entropy-Aware Speculative Decoding (EASD)

---

**Require:** Context $s$, target model $\text{LLM}_{\text{targ}}$, draft model $\text{LLM}_{\text{draft}}$, top-$n$ size $n$, thresholds $\tau_H, \tau_O$
**Ensure:** Generated token sequence
1: $t \leftarrow 1, output \leftarrow [\,]$
2: **while** not end of sequence **do**
3:     Draft model generates $M$ candidates:
4:        $(c_1, P_d(1)), \ldots, (c_M, P_d(M)) = \text{LLM}_{\text{draft}}^{(M)}(s)$
5:     Target model computes distributions:
6:        $(P_t(1), \ldots, P_t(M+1)) = \text{LLM}_{\text{targ}}(c_1, \ldots, c_M; s)$
7:     **for** $i = 1$ **to** $M$ **do**
8:        Compute entropy for both models:
9:           $H_d^{(i)} = -\sum_{v \in \mathcal{V}} P_d^{(i)}(v) \log P_d^{(i)}(v)$
10:          $H_t^{(i)} = -\sum_{v \in \mathcal{V}} P_t^{(i)}(v) \log P_t^{(i)}(v)$
11:        Compute top-$n$ overlap:
12:          overlap $\leftarrow |T_d^n \cap T_t^n|/n$
13:        **if** $H_d^{(i)} > \tau_H$ **and** $H_t^{(i)} > \tau_H$ **and** overlap $> \tau_O$ **then**
14:           Apply dynamic penalty: $P_t^{(i)}(c_i) \leftarrow 0$
15:           Renormalize: $P_t^{(i)}(v) \leftarrow P_t^{(i)}(v)/\sum_{v \neq c_i} P_t^{(i)}(v), \forall v \neq c_i$
16:        **end if**
17:        Sample $\epsilon_i \sim \mathcal{U}(0, 1)$
18:        **if** $\epsilon_i \leq P_t^{(i)}(c_i)/P_d^{(i)}(c_i)$ **then**
19:          $output \leftarrow output \oplus c_i$ {Accept token}
20:        **else**
21:          Sample token $c_i \sim P_t^{(i)}$ directly
22:          $output \leftarrow output \oplus c_i$
23:          **break** {Stop draft sequence verification}
24:        **end if**
25:     **end for**
26:     $t \leftarrow t + 1$
27: **end while**
28: **return** $output$

---

# C  CASE STUDY

## C.1  SUCCESS CASE

**GPQA Case**: The discrepancy between the original answer and the EASD-enhanced answer stems from the first decisive token used to identify the initial reactant. For the first reaction,

$$A + H_2SO_4 \rightarrow 2,8\text{-dimethylspiro}[4.5]\text{decan} - 6\text{one},$$

the original answer begins with the token `an alcohol`, whereas the EASD-enhanced answer starts with `a diol`. Although this difference may appear subtle, the choice of article (a vs. an) immediately commits the model to distinct classes of candidate structures. In the original case, `an alcohol` frames the reasoning around monohydroxyl precursors, leading subsequent steps to focus on single-hydroxyl compounds that are unlikely to yield the target spiro ketone. By contrast, the EASD-generated token `a diol` emphasizes a dihydroxyl precursor, guiding the reasoning toward structures capable of undergoing the necessary dehydration to form the spiro ring.

This initial token choice effectively anchors the downstream reasoning: it determines which molecular candidates are explored, which transformations are deemed plausible, and ultimately which reactant is selected. By avoiding early misdirection, the EASD-generated token ensures that subsequent inference steps remain consistent with the chemical requirements of the reaction.

In summary, the success of EASD in this case can be attributed to generating the more appropriate initial token (`a diol`), which not only steers the model toward chemically viable candidates but also prevents error propagation from the outset, thereby enabling the correct identification of the reactant.

To solve this problem, we need to identify the reactants (A and B) for each reaction given the products. For the first reaction:

$A+H2SO4 \rightarrow 2,8$–dimethylspiro[4.5]decan–6one

The product is a spiro compound, which suggests that the reactant A could be an alcohol that undergoes dehydration to form the ketone. The structure of the product indicates that the reactant A should be **a diol that forms a spiro ring upon dehydration**. Among the options provided, **2,7-dimethyloctahydronaphthalene-4a,8a-diol** can form a spiro compound **upon** dehydration, making it a **plausible** candidate for A. For the second reaction:

$B+BuLi+H+ \rightarrow 4$–methyl–1–phenylpent–3en–1ol

This reaction involves the use of butyllithium (BuLi), which is **often** used in organic synthesis to form carbon-carbon bonds, particularly in Wittig-like reactions or in the formation of alkenes from carbonyl compounds. The product is an alcohol **with** a double bond, suggesting that B **might** be a carbonyl compound that reacts with BuLi to form an alkoxide intermediate, which then protonates to form the alcohol. **The** structure of the product suggests **that** B could be a ketone or an ester that can **form** the alcohol upon reduction. Among the options provided, 4-methyl-1-phenylpent-3-en-1-one is a ketone that can form the alcohol upon reduction, making it a plausible candidate for B. **Thus**, the **correct** choice is: \boxed{A}

To solve this problem, we need to identify the reactants (A and B) for each reaction given the products. For the first reaction:

$A+H2SO4 \rightarrow 2,8$–dimethylspiro[4.5]decan–6one

The product is a spiro compound, which suggests that the reactant A could be an alcohol that undergoes dehydration to form the ketone. The structure of the product indicates that the reactant **A should be an alcohol with a similar backbone but with an -OH group instead of the ketone.** Therefore, the most likely candidate for A is **2,8–dimethylspiro[4.5]decan–6ol** . For the second reaction:

$B+BuLi+H+ \rightarrow 4$–methyl–1–phenylpent–3en–1ol

This reaction involves the use of a strong base (BuLi) followed by protonation (H+). This sequence of steps is typical for the formation of alkenes from alcohols or ethers through elimination reactions. Given the product is an alcohol, it suggests that B might be an ether or an alcohol that can undergo such a transformation. The structure of the product suggests that B could be an ether with a similar backbone, specifically ((3–methylbut–2en–1–yl)oxy)methylbenzene . Therefore, the correct choice is: \boxed{C}

Figure 2: **Success Case:** Answer comparison on a GPQA reasoning problem, illustrating the difference between the original output and the EASD-enhanced output.

**Math Case**: The primary divergence between the original answer and the EASD-enhanced answer occurs at the first token following the transition word `However,`. In the original answer, this token is `because`, whereas in the EASD-enhanced answer it is `since`. While both tokens introduce a causal explanation, this subtle difference exerts a significant influence on the subsequent reasoning trajectory.

In the original answer, beginning with `because` directs the model's reasoning toward exploring the maximal range of values for each variable, which ultimately leads to an overestimation of the number of valid triples as $n+1$. By contrast, the EASD-enhanced answer, beginning with `since`, subtly reorients the reasoning toward identifying the minimal number of distinct values necessary to construct valid triples. This shift in focus ensures that the model considers only feasible combinations, effectively guiding it to the correct maximal number of triples, $N(n) = 3$.

Thus, the success of EASD in this case can be attributed to the generation of a more precise initial causal token. By influencing the early steps of reasoning, EASD prevents error propagation, aligns the model's inference with the problem constraints, and ultimately leads to the correct conclusion.

To determine $N(n)$ for all $n \geq 2$, we need to find the maximum number of triples $(a_i, b_i, c_i)$ such that each triple satisfies $a_i + b_i + c_i = n$ and no two triples share the same value in any position (i.e., if $i \neq j$, then $a_i \neq a_j$, $b_i \neq b_j$, and $c_i \neq c_j$). First, consider the possible values for $a_i, b_i$, and $c_i$. Since they are nonnegative integers and their sum is $n$, each of them can take any value from 0 to $n$. However, since no two triples can have the same value in any position, the number of distinct values that can be used for $a_i, b_i$, and $c_i$ is limited to the number of available distinct values, which is $n+1$ (the integers from 0 to $n$). Since we need three distinct values for each triple and these values must be different across all triples, the maximum number of such triples is limited by the number of ways we can choose 3 distinct values out of $n+1$ values. This means we can have at most $\min(n+1, 3)$ distinct values per position, but since we need three distinct values for each triple, the limiting factor is the smallest set of distinct values we can use, which is 3 when $n \geq 2$. Therefore, the maximum number of such triples is 3, because we can form exactly 3 triples using the values 0, 1, and $n-1$ in some permutation. For example, if $n=3$, the triples could be (0,1,2), (1,2,0), and (2,0,1). Thus, the maximal number of such triples $N(n)$ is 3 for all $n \geq 2$. The final answer is: 3

To determine $N(n)$ for all $n \geq 2$, we need to find the maximum number of triples $(a_i, b_i, c_i)$ such that each triple satisfies $a_i + b_i + c_i = n$ and no two triples share the same value in any position (i.e., if $i \neq j$, then $a_i \neq a_j$, $b_i \neq b_j$, and $c_i \neq c_j$). First, consider the possible values for $a_i, b_i$, and $c_i$. Since they are nonnegative integers and their sum is $n$, each of them can take any value from 0 to $n$. However, because no two triples can have the same value in any position, the number of distinct values each variable can take is limited by the number of triples. Let's denote the number of triples by $k$. Then, each of $a_i, b_i$, and $c_i$ must take $k$ different values. The smallest possible value for each variable is 0 and the largest possible value is $n$. Therefore, the number of distinct values each variable can take is at most $n+1$. Since we need $k$ distinct values for each of the three variables, we have $k \leq n+1$. However, since $a_i + b_i + c_i = n$, the maximum value for any one of the variables is $n$, and the minimum value is 0. This means that if we have $k$ distinct values, the largest value must be $n$ and the smallest value must be 0, which implies that the values must be $0, 1, 2, ..., k-1$. For these values to sum to $n$, the largest value $k-1$ must be $n$, so $k-1 = n$ or $k = n+1$. We need to check if it is possible to form $n+1$ such triples. Consider the following construction: $(0,1,n-1), (1,2,n-2), (2,3,n-3), ..., (n-1,n,0)$. This gives us $n+1$ triples where each variable takes on a distinct value from 0 to $z$. It is easy to verify that this construction satisfies both conditions: the sum of each triple is $n$, and no two triples share the same value in any position. Therefore, the maximal number of such triples is $n+1$. Thus, the answer is: $n+1$

Figure 3: **Success Case:** Answer comparison on a Math reasoning problem, illustrating the difference between the original output and the EASD-output.

## C.2 BAD CASE

Not all cases can be successful. In this bad case, the EASD-enhanced model produced an incorrect answer due to subtle token-level interactions. Specifically, the consecutive appearance of the tokens `since` and `only` influenced the model's reasoning path. Instead of correctly applying the inclusion-exclusion principle and accounting for overlaps between multiples of 10 and 15, the model mistakenly subtracted only the multiples of 10 from the total count of multiples of 3 or 5. This led to an overcount and resulted in the answer 110, rather than the correct 100.

The error illustrates how token selection, even when seemingly minor, can propagate through subsequent reasoning steps and significantly affect the final outcome. In this case, the model's focus on the word `only` caused it to overlook necessary adjustments for overlapping sets, demonstrating that EASD's strategy of optimizing initial tokens does not guarantee error-free reasoning in all combinatorial or highly structured problems.

## D SPEED COMPARISON

We conducted a detailed efficiency analysis of EASD under the setting where the draft model is 7B and the target model is 32B, using the hyperparameters applied in our main experiments (Entropy = 2, Top-n = 0.8). Table 5 reports the token generation speed (tokens per second, tok/s) for the Single Model baseline, EASD, and standard speculative decoding (SD) across six reasoning datasets: Olympia, Minerva, Math500, AMC23, Aime24, and GPQA. We also report the relative speedup compared to the single model baseline.

From the results, we observe that EASD achieves an average token generation speed of 17 tok/s across the datasets, which is slightly lower than SD at 18 tok/s. This minor slowdown is expected, as EASD introduces two additional conditions—the high-entropy penalty and top-n overlap check during decoding. Nevertheless, EASD still achieves significant speedup relative to the single model baseline, confirming that most tokens are efficiently accepted from the draft model without invoking the target model.

To solve this problem, we will use the principle of inclusion-exclusion and then subtract the numbers that are multiples of 10 or 15. **Step 1: Counting Multiples of 3 and 5** First, let's count how many numbers up to 300 are multiples of 3 or 5. - The number of multiples of 3 up to 300 is $\lfloor 300/3 \rfloor = 100$
. - The number of multiples of 5 up to 300 is $\lfloor 300/5 \rfloor = 60$
. - The number of multiples of both 3 and 5 (i.e., multiples of 15) up to 300 is $\lfloor 300/15 \rfloor = 20$
. Using the principle of inclusion-exclusion, the total number of multiples of 3 or 5 up to 300 is:
$100 + 60 - 20 = 140$
**Step 2: Subtracting Multiples of 10 and 15** Next, we need to exclude the numbers that are multiples of 10 or 15. - The number of multiples of 10 up to 300 is $\lfloor 300/10 \rfloor = 30$
. - The number of multiples of 15 up to 300 is already calculated as 20. Since every multiple of 15 is also a multiple of 5, we only need to subtract the multiples of 10 from our previous count because they are not already excluded in the initial calculation. Thus, the number of multiples of 10 that are also multiples of 3 or 5 is 30. Therefore, the number of positive integers less than or equal to 300 that are multiples of 3 or 5 but are not multiples of 10 or 15 is:
$140 - 30 = 110$

So, the final answer is: 110.

To solve this problem, we will use the principle of inclusion-exclusion and then subtract the numbers that are multiples of 10 or 15. **Step 1: Counting Multiples of 3 and 5** First, let's count how many numbers up to 300 are multiples of 3 or 5. - The number of multiples of 3 up to 300 is $\lfloor 300/3 \rfloor = 100$
. - The number of multiples of 5 up to 300 is $\lfloor 300/5 \rfloor = 60$
. - The number of multiples of both 3 and 5 (i.e., multiples of 15) up to 300 is $\lfloor 300/15 \rfloor = 20$
. Using the principle of inclusion-exclusion, the total number of multiples of 3 or 5 up to 300 is:
$100 + 60 - 20 = 140$
**Step 2: Subtracting Multiples of 10 and 15** Next, we need to exclude the numbers that are multiples of 10 or 15 from our previous count. - The number of multiples of 10 up to 300 is $\lfloor 300/10 \rfloor = 30$
. - The number of multiples of 15 up to 300 is already calculated as 20. However, we must be careful not to double-count the multiples of 30 (since they are counted in both multiples of 10 and 15). - The number of multiples of 30 up to 300 is $\lfloor 300/30 \rfloor = 10$. So, the number of multiples of 10 or 15 up to 300 is:
$30 + 20 - 10 = 40$
**Step 3: Final Calculation** Subtract the multiples of 10 or 15 from the total multiples of 3 or 5:
$140 - 40 = 100$
Therefore, the number of positive integers less than or equal to 300 that are multiples of 3 or 5, but are not multiples of 10 or 15 is 100.

Figure 4: **Bad Case:** Answer comparison on a Math reasoning problem, illustrating the difference between the original output and the EASD-output.

Examining dataset-specific results, we find that on most math reasoning datasets, EASD achieves a token speed of 17–19 tok/s, corresponding to 1.21–1.36× speedup compared to the single model. On GPQA, a general reasoning dataset, the speed is slightly lower, reflecting a higher fraction of tokens being penalized by EASD due to increased uncertainty. These results indicate that while the additional token-level checks introduce a modest computational overhead, EASD maintains near-SD-level efficiency across diverse datasets.

Table 5: Token generation speed (tokens per second) and speedup of EASD and SD compared to the single model baseline across multiple datasets.

| Dataset | Single Model tok/s | EASD tok/s | Speedup | SD tok/s | Speedup |
|---------|---------|---------|---------|---------|---------|
| Olympia | 14 | 19 | 1.36 | 19 | 1.36 |
| Minerva | 14 | 17 | 1.21 | 18 | 1.29 |
| Math500 | 14 | 19 | 1.36 | 20 | 1.43 |
| AMC23 | 14 | 19 | 1.36 | 18 | 1.29 |
| Aime24 | 14 | 17 | 1.21 | 17 | 1.21 |
| GPQA | 14 | 13 | 0.93 | 14 | 1 |
| Avg | 14 | 17 | 1.21 | 18 | 1.29 |

## E  HYPERPARAMETER SENSITIVITY STUDY

We conducted a comprehensive hyperparameter study to evaluate the impact of varying the entropy and top-n thresholds on EASD's performance, focusing on both token penalization and output accuracy. Table 6 reports the results across six reasoning datasets for a range of entropy thresholds (1, 1.5, 2) and top-n thresholds (0, 0.2, 0.4, 0.6, 0.8, 1).

The results reveal several key trends. First, increasing the thresholds to penalize less tokens generally leads to improved output accuracy, confirming that actively rejecting low-confidence tokens enhances reasoning quality. Second, optimal thresholds vary slightly across datasets: Aime24 and

GPQA achieve the best average accuracy with Entropy = 2 and Top-n = 0.8, whereas AMC23 benefits from slightly lower thresholds.

These findings indicate that while some dataset-specific tuning can further improve performance, the default values used in our experiments (Entropy = 2, Top-n = 0.8) offer a robust trade-off between efficiency and output quality across diverse tasks. Overall, this analysis demonstrates that EASD is effective and practically tunable, even when validation data are limited or unavailable.

Table 6: EASD performance under different entropy and top-n threshold settings (%).

| Entropy | Top-n | Olympia | Minerva | Math500 | AMC23 | Aime24 | GPQA | Avg |
|---------|-------|---------|---------|---------|-------|--------|------|-----|
| 1 | | 45.33 | 46.69 | 80.4 | 67.5 | 13.33 | 43.94 | 49.53 |
| 1.5 | 0 | 46.22 | 44.85 | 82.4 | 62.5 | 13.33 | 49.49 | 49.80 |
| 2 | | 47.41 | 44.85 | 80.8 | 65 | 20 | 52.53 | 51.77 |
| 1 | | 45.33 | 46.69 | 80.4 | 67.5 | 13.33 | 43.94 | 49.53 |
| 1.5 | 0.2 | 46.22 | 44.85 | 82.4 | 62.5 | 13.33 | 49.49 | 49.80 |
| 2 | | 47.41 | 44.85 | 80.8 | 65 | 20 | 52.53 | 51.77 |
| 1 | | 45.33 | 47.06 | 80.2 | 65 | 13.33 | 44.44 | 49.23 |
| 1.5 | 0.4 | 46.22 | 44.85 | **82.4** | 62.5 | 13.33 | 50.51 | 49.97 |
| 2 | | 47.41 | 44.12 | 80.6 | 65 | 20 | 54.04 | 51.86 |
| 1 | | 45.04 | **48.16** | 79.2 | 67.5 | 13.33 | 45.45 | 49.78 |
| 1.5 | 0.6 | 46.37 | 45.59 | 82 | 62.5 | 13.33 | 52.02 | 50.30 |
| 2 | | **48.3** | 45.59 | 80.8 | 65 | 20 | 50.51 | 51.70 |
| 1 | | 44.89 | 47.06 | 81.6 | 67.5 | 16.67 | 46.46 | 50.70 |
| 1.5 | **0.8** | 45.33 | 48.16 | 82 | 62.5 | 20 | 51.52 | 51.59 |
| **2** | | 48.15 | 47.06 | 80.8 | 62.5 | **23.33** | **55.55** | **52.89** |
| 1 | | 46.52 | 44.85 | 79.2 | **77.5** | 16.67 | 46.47 | 51.87 |
| 1.5 | 1 | 47.41 | 43.75 | 81.4 | 67.5 | 16.67 | 47.48 | 50.70 |
| 2 | | 46.81 | 46.32 | 81.2 | 62.5 | 20 | 49.49 | 51.05 |

## F  PENALIZED TOKEN FREQUENCY ANALYSIS

To further understand the effect of EASD's penalty mechanism, we analyzed how frequently tokens are penalized under varying entropy and top-n thresholds, as reported in Table 7. The analysis reveals clear trends: higher entropy or top-n thresholds reduce the number of penalized tokens, while lower thresholds increase token penalization. This behavior is consistent across different datasets and demonstrates that the dual-condition mechanism effectively controls how aggressively the model intervenes in token generation.

Interestingly, our experiments show that when fewer tokens are penalized, the overall accuracy of the generated outputs tends to be higher. This suggests that EASD is able to focus its corrections on the most uncertain or high-impact tokens, avoiding unnecessary disruption of tokens that are already well-predicted by the draft model. By selectively penalizing a targeted subset of tokens, EASD achieves a balance between efficiency and output quality, ensuring that the applied corrections meaningfully improve reasoning performance without introducing excessive noise or computational overhead.

Table 7: Frequency of penalized tokens under different entropy and top-n threshold settings (%).

| Entropy | Top-n | Olympia | Minerva | Math500 | AMC23 | Aime24 | GPQA | Avg |
|---------|-------|---------|---------|---------|-------|--------|-------|-------|
| 1 | | 7.69 | 8.98 | 6.97 | 8.05 | 9.85 | 26.34 | 11.31 |
| 1.5 | 0 | 3.05 | 4.18 | 2.7 | 3.38 | 4.91 | 15.1 | 5.55 |
| 2 | | 1.11 | 1.71 | 0.91 | 1.3 | 1.57 | 8.47 | 2.51 |
| 1 | | 7.69 | 8.98 | 6.97 | 8.05 | 9.85 | 26.34 | 11.31 |
| 1.5 | 0.2 | 3.05 | 4.18 | 2.7 | 3.38 | 4.91 | 15.1 | 5.55 |
| 2 | | 1.11 | 1.71 | 0.91 | 1.3 | 1.57 | 8.47 | 2.51 |
| 1 | | 7.65 | 8.85 | 6.99 | 7.98 | 9.97 | 25.87 | 11.22 |
| 1.5 | 0.4 | 3.03 | 4.14 | 2.7 | 3.37 | 4.91 | 15.03 | 5.53 |
| 2 | | 1.1 | 1.67 | 0.91 | 1.29 | 1.57 | 8.73 | 2.55 |
| 1 | | 7.42 | 8.61 | 6.66 | 7.69 | 9.32 | 22.96 | 10.44 |
| 1.5 | 0.6 | 2.89 | 3.83 | 2.62 | 3.14 | 4.37 | 13.51 | 5.06 |
| 2 | | 1.01 | 1.49 | 0.87 | 1.18 | 1.43 | 6.94 | 2.15 |
| 1 | | 5.79 | 6.73 | 5.67 | 6.13 | 7.53 | 14.28 | 7.69 |
| 1.5 | **0.8** | 2.16 | 3 | 2.02 | 2.34 | 3.05 | 8.11 | 3.45 |
| **2** | | 0.7 | 0.97 | 0.63 | 0.77 | 0.97 | 3.97 | 1.34 |
| 1 | | 1.98 | 2.4 | 2.13 | 1.94 | 2.61 | 3.38 | 2.41 |
| 1.5 | 1 | 0.7 | 0.94 | 0.77 | 0.73 | 0.87 | 1.81 | 0.97 |
| 2 | | 0.18 | 0.26 | 0.19 | 0.19 | 0.22 | 0.72 | 0.29 |

