# OpenReview forum: "Entropy-Aware Speculative Decoding Toward Improved LLM Reasoning"
_ICLR.cc/2026/Conference — Submitted to ICLR 2026_

### Official Review · Reviewer_D3Fk · 2025-10-20

**Soundness:** 1
**Presentation:** 2
**Contribution:** 1
**Rating:** 2
**Confidence:** 4

**Summary:**

This paper proposes Entropy-Aware Speculative Decoding, a training-free method designed to enhance standard speculative decoding by introducing an entropy-based penalty mechanism. At each decoding step, the method measures both the draft and target model entropies to decide whether to accept or reject a token. The aim is to prevent low-confidence tokens from propagating and potentially improve accuracy. Experimental results on multiple reasoning benchmarks indicate that EASD achieves higher accuracy than standard SD and even outperforms the target model alone in some cases.

**Strengths:**

Simple and training-free: The proposed method does not require additional training, which increases its potential applicability in practice.

**Weaknesses:**

1. Lack of actual speed measurement: Despite being a speculative decoding paper, there are no end-to-end latency or throughput experiments. The “Computation Analysis” section only reports average generated token counts as a proxy for computational cost, which is insufficient. Speculative decoding methods are typically evaluated by real speedups (e.g., wall-clock latency or tokens/s) on standard hardware configurations. This omission significantly undermines the main claim regarding efficiency.

2. Weak performance on the accuracy results: The vanilla SD methods are lossless, which means that the gap between SD and Single Model in Table 1 should be considered as the experiment error. The gap between EASD and Single Model are not significantly higher than this experimental error, which weakens the significance of the experiments in Table 1.

3. Missing ablation on critical hyperparameters: The paper mentions determining the entropy threshold using the LIMO dataset but does not provide analysis of sensitivity to this choice. It is unclear how robust the method is to different thresholds or entropy percentile values.

**Questions:**

1. Why is the wallclock speedup not included in Computation Analysis part?

2. Can you provide more detailed parameter sensitivity analysis?

---

> ### Author Response · Authors · 2025-12-03
> **Response to Reviewer D3Fk -1**
>
> We sincerely thank you for your careful evaluation and constructive feedback.We provide detailed responses below to address the specific concerns raised.
>
> `Weakness2: Weak performance on the accuracy results: The vanilla SD methods are lossless, which means that the gap between SD and Single Model in Table 1 should be considered as the experiment error. The gap between EASD and Single Model are not significantly higher than this experimental error, which weakens the significance of the experiments in Table 1.`
>
> Answer: We thank you for this comment. While SD is theoretically unbiased and guarantees accuracy equal to the target model, it often underperforms in practice, as observed in prior work (Chen et al., 2023; Liao et al., 2025). Therefore, the gaps reported in Table 1 between EASD and the Single Model reflect genuine performance improvements in practical settings. EASD consistently enhances output quality, demonstrating benefits beyond what could be attributed to experimental noise.
>
>
> [1] Chen, Charlie, et al. "Accelerating large language model decoding with speculative sampling." arXiv preprint arXiv:2302.01318 (2023).
>
> [2] Liao, Baohao, et al. "Reward-Guided Speculative Decoding for Efficient LLM Reasoning." Forty-second International Conference on Machine Learning(2025).
>
> `Question1: Why is the wallclock speedup not included in Computation Analysis part?`
>
> Answer: We thank you for this question. We conducted detailed speed measurements under the setting where the draft model is 7B and the target model is 32B, using the hyperparameters in our paper (Entropy=2, Top-n=0.8). We measured token generation speed for EASD, SD, and Single Model. The results show that EASD is slightly slower than SD, as expected due to the additional conditions, but the overall efficiency remains comparable. We have now clarified these speed measurements in the revised manuscript to provide a more complete and reproducible computational analysis.
>
> Tabel 5. Token generation speed (tokens per second) and speedup of EASD and SD compared to the single model baseline across multiple datasets.
> | Dataset | Single Model tok/s | EASD tok/s | Speedup | SD tok/s | Speedup |
> | :-----: | :----------------: | :--------: | :-----: | :------: | :-----: |
> | Olympia |         14         |     19     |  1.36   |    19    |  1.36   |
> | Minerva |         14         |     17     |  1.21   |    18    |  1.29   |
> | Math500 |         14         |     19     |  1.36   |    20    |  1.43   |
> |  AMC23  |         14         |     19     |  1.36   |    18    |  1.29   |
> | AIME24  |         14         |     17     |  1.21   |    17    |  1.21   |
> |  GPQA   |         14         |     13     |  0.93   |    14    |    1    |
> |   Avg   |         14         |     17     |  1.21   |    18    |  1.29   |

---

> ### Author Response · Authors · 2025-12-03
> **Response to Reviewer D3Fk -2**
>
> `Question2: Can you provide more detailed parameter sensitivity analysis?`
>
> Answer: We thank you for this question. To provide a more detailed parameter sensitivity analysis, we conducted hyperparameter studies under the setting where the draft model is 7B and the target model is 32B. Our experiments vary the entropy and top-n thresholds and measure their effects on token penalization and output accuracy across different datasets. The results show that:
>
> - As the entropy threshold increases, the number of tokens penalized by EASD decreases, while the accuracy of generated outputs shows a slight upward trend.
>
> - Different datasets exhibit slightly different optimal thresholds for achieving the highest accuracy, indicating that dataset-specific tuning can further improve performance.
>
> - Moderate values (e.g., Entropy=2, Top-n=0.8, used in our main experiments) provide a reasonable balance between token penalties, accuracy, and efficiency.
>
> These observations provide practical guidance for threshold selection when validation data are unavailable. We have added these analyses and guidelines to the revised manuscript to give a more comprehensive view of parameter sensitivity.
>
> Tabel 6. EASD performance under different entropy and top-n threshold settings (\%).
> | Entropy |  Top-n  | Olympia  |  Minerva  | Math500  |  AMC23   |  Aime24   |   GPQA    |    Avg    |
> | :-----: | :-----: | :------: | :-------: | :------: | :------: | :-------: | :-------: | :-------: |
> |    1    |    0    |  45.33   |   46.69   |   80.4   |   67.5   |   13.33   |   43.94   |   49.53   |
> |   1.5   |    0    |  46.22   |   44.85   |   82.4   |   62.5   |   13.33   |   49.49   |   49.80   |
> |    2    |    0    |  47.41   |   44.85   |   80.8   |    65    |    20     |   52.53   |   51.77   |
> |    1    |   0.2   |  45.33   |   46.69   |   80.4   |   67.5   |   13.33   |   43.94   |   49.53   |
> |   1.5   |   0.2   |  46.22   |   44.85   |   82.4   |   62.5   |   13.33   |   49.49   |   49.80   |
> |    2    |   0.2   |  47.41   |   44.85   |   80.8   |    65    |    20     |   52.53   |   51.77   |
> |    1    |   0.4   |  45.33   |   47.06   |   80.2   |    65    |   13.33   |   44.44   |   49.23   |
> |   1.5   |   0.4   |  46.22   |   44.85   | **82.4** |   62.5   |   13.33   |   50.51   |   49.97   |
> |    2    |   0.4   |  47.41   |   44.12   |   80.6   |    65    |    20     |   54.04   |   51.86   |
> |    1    |   0.6   |  45.04   | **48.16** |   79.2   |   67.5   |   13.33   |   45.45   |   49.78   |
> |   1.5   |   0.6   |  46.37   |   45.59   |    82    |   62.5   |   13.33   |   52.02   |   50.30   |
> |    2    |   0.6   | **48.3** |   45.59   |   80.8   |    65    |    20     |   50.51   |   51.70   |
> |    1    |   0.8   |  44.89   |   47.06   |   81.6   |   67.5   |   16.67   |   46.46   |   50.70   |
> |   1.5   |   0.8   |  45.33   |   48.16   |    82    |   62.5   |    20     |   51.52   |   51.59   |
> |  **2**  | **0.8** |  48.15   |   47.06   |   80.8   |   62.5   | **23.33** | **55.55** | **52.89** |
> |    1    |    1    |  46.52   |   44.85   |   79.2   | **77.5** |   16.67   |   46.47   |   51.87   |
> |   1.5   |    1    |  47.41   |   43.75   |   81.4   |   67.5   |   16.67   |   47.48   |   50.70   |
> |    2    |    1    |  46.81   |   46.32   |   81.2   |   62.5   |    20     |   49.49   |   51.05   |

---

### Official Review · Reviewer_x1Uw · 2025-10-22

**Soundness:** 2
**Presentation:** 2
**Contribution:** 2
**Rating:** 2
**Confidence:** 4

**Summary:**

This paper proposes EASD, a training-free algorithm for speculative decoding in LLM reasoning. The two key components incorporate entropy-based uncertainty and distributional overlap (top-n) between the draft and target models to dynamically regulate token acceptance. When both models exhibit high entropy and their top-N token predictions substantially overlap, EASD penalizes the target model’s probability for the draft token and forces resampling, preventing low-confidence tokens from propagating errors. Experiments are conducted on math reasoning datasets and GPQA, showing that EASD outperforms speculative decoding baselines. Further analyses include an ablation study, computational analysis, and case studies, which help better understand the method.

**Strengths:**

- The idea of EASD is simple and effective: it is conceptually clear and a training-free extension to speculative decoding that leverages entropy and distributional overlap to improve reasoning quality, showing improvement in performance without much increasing complexity or computational cost.
- EASD consistently outperforms both standard and reward-guided speculative decoding (RSD) across diverse reasoning benchmarks.
- The paper is easy to read and understand, with analyses help interpret the performance and design of the method.

**Weaknesses:**

1. For methodology design, while leveraging entropy and top-n is a signal, the choice of the threshold is pretty heuristic. For entropy, the threshold has to be pre-defined through computation on a validation set, while for top-n, it is pre-defined by a heuristic. This reliance on fixed thresholds raises concerns about the method’s robustness and generalizability across different model pairs or domains. Since entropy distributions and token overlap patterns can vary significantly between architectures or tasks, the optimal thresholds for triggering the dynamic penalty may need to be re-tuned for each new setting.
2. I am concerned with the comprehensiveness of the main experiments and additional analyses. For example, the experiments are only done using Qwen-2.5-Series, and the experiments are mostly centered around math reasoning. To further claim general LLM reasoning, I would suggest the authors add experiments on code reasoning and more general ones, such as HumanEval and MMLU.
3. For speculative decoding, IMO, algorithms are designed for better efficiency claims. Instead of mainly claiming performance improvement, additional analyses on efficiency, in addition to Section 4.4 would be appreciated.
4. The performance improvement is not significant, while EASD will incur additional computation overhead. The authors should show strong evidence that the tradeoff between effectiveness and efficiency is worthwhile.

**Questions:**

1. For entropy and top-n threshold selection, how to set an appropriate value for different datasets and methods? Providing guidance would be better.
2. Could you provide a more comprehensive analysis for efficiency? E.g., a time breakdown to draft/verification, tokens per second, etc.
3. IMO, Figure 1 is not easy to understand. Removing baselines and leaving only EASD, with clearer illustrations on tokens would be better.

---

> ### Author Response · Authors · 2025-12-03
> **Response to Reviewer x1Uw -1**
>
> We sincerely thank you for your careful evaluation and detailed feedback. We provide detailed responses below to address the specific questions and concerns.
>
> `Question1: For entropy and top-n threshold selection, how to set an appropriate value for different datasets and methods? Providing guidance would be better.`
>
> Answer: We thank you for this question. In our experiments reported in the paper, we used the same entropy and top-n thresholds across different datasets. However, we acknowledge that different datasets can exhibit slightly different behavior under varying threshold settings. To investigate this, we conducted a hyperparameter study with the draft model at 7B and the target model at 32B. The results show that the threshold values that achieve the highest accuracy vary slightly across datasets. These findings suggest that, while our default settings (Entropy=2, Top-n=0.8) provide a reasonable starting point, slight tuning may further improve performance on specific datasets.
>
> Tabel 6. EASD performance under different entropy and top-n threshold settings (\%).
> | Entropy |  Top-n  | Olympia  |  Minerva  | Math500  |  AMC23   |  Aime24   |   GPQA    |    Avg    |
> | :-----: | :-----: | :------: | :-------: | :------: | :------: | :-------: | :-------: | :-------: |
> |    1    |    0    |  45.33   |   46.69   |   80.4   |   67.5   |   13.33   |   43.94   |   49.53   |
> |   1.5   |    0    |  46.22   |   44.85   |   82.4   |   62.5   |   13.33   |   49.49   |   49.80   |
> |    2    |    0    |  47.41   |   44.85   |   80.8   |    65    |    20     |   52.53   |   51.77   |
> |    1    |   0.2   |  45.33   |   46.69   |   80.4   |   67.5   |   13.33   |   43.94   |   49.53   |
> |   1.5   |   0.2   |  46.22   |   44.85   |   82.4   |   62.5   |   13.33   |   49.49   |   49.80   |
> |    2    |   0.2   |  47.41   |   44.85   |   80.8   |    65    |    20     |   52.53   |   51.77   |
> |    1    |   0.4   |  45.33   |   47.06   |   80.2   |    65    |   13.33   |   44.44   |   49.23   |
> |   1.5   |   0.4   |  46.22   |   44.85   | **82.4** |   62.5   |   13.33   |   50.51   |   49.97   |
> |    2    |   0.4   |  47.41   |   44.12   |   80.6   |    65    |    20     |   54.04   |   51.86   |
> |    1    |   0.6   |  45.04   | **48.16** |   79.2   |   67.5   |   13.33   |   45.45   |   49.78   |
> |   1.5   |   0.6   |  46.37   |   45.59   |    82    |   62.5   |   13.33   |   52.02   |   50.30   |
> |    2    |   0.6   | **48.3** |   45.59   |   80.8   |    65    |    20     |   50.51   |   51.70   |
> |    1    |   0.8   |  44.89   |   47.06   |   81.6   |   67.5   |   16.67   |   46.46   |   50.70   |
> |   1.5   |   0.8   |  45.33   |   48.16   |    82    |   62.5   |    20     |   51.52   |   51.59   |
> |  **2**  | **0.8** |  48.15   |   47.06   |   80.8   |   62.5   | **23.33** | **55.55** | **52.89** |
> |    1    |    1    |  46.52   |   44.85   |   79.2   | **77.5** |   16.67   |   46.47   |   51.87   |
> |   1.5   |    1    |  47.41   |   43.75   |   81.4   |   67.5   |   16.67   |   47.48   |   50.70   |
> |    2    |    1    |  46.81   |   46.32   |   81.2   |   62.5   |    20     |   49.49   |   51.05   |

---

> ### Author Response · Authors · 2025-12-03
> **Response to Reviewer x1Uw -2**
>
> `Question2: Could you provide a more comprehensive analysis for efficiency? E.g., a time breakdown to draft/verification, tokens per second, etc.`
>
> Answer: We thank you for this suggestion. To provide a more comprehensive analysis of efficiency, we conducted experiments with the draft model at 7B and the target model at 32B, using the hyperparameters in our paper (Entropy=2, Top-n=0.8). We measured the token generation speed for EASD, SD, and Single Model, and observed that EASD is slightly slower than SD, as expected due to the additional high-entropy and top-n overlap conditions.
>
> Tabel 5. Token generation speed (tokens per second) and speedup of EASD and SD compared to the single model baseline across multiple datasets.
> | Dataset | Single Model tok/s | EASD tok/s | Speedup | SD tok/s | Speedup |
> | :-----: | :----------------: | :--------: | :-----: | :------: | :-----: |
> | Olympia |         14         |     19     |  1.36   |    19    |  1.36   |
> | Minerva |         14         |     17     |  1.21   |    18    |  1.29   |
> | Math500 |         14         |     19     |  1.36   |    20    |  1.43   |
> |  AMC23  |         14         |     19     |  1.36   |    18    |  1.29   |
> | AIME24  |         14         |     17     |  1.21   |    17    |  1.21   |
> |  GPQA   |         14         |     13     |  0.93   |    14    |    1    |
> |   Avg   |         14         |     17     |  1.21   |    18    |  1.29   |
>
> `Question3: IMO, Figure 1 is not easy to understand. Removing baselines and leaving only EASD, with clearer illustrations on tokens would be better.`
>
> Answer: We thank you for this suggestion. Figure 1 illustrates the workflow of EASD, showing how tokens are processed through the draft model, evaluated with the high-entropy and top-n overlap conditions, and potentially penalized. The inclusion of baseline methods is intended to highlight the practical improvements and differences achieved by EASD compared to existing approaches. We plan to further optimize the figure in future versions for enhanced clarity, potentially focusing on EASD and providing more detailed token-level illustrations.

---

### Official Review · Reviewer_AVTk · 2025-10-25

**Soundness:** 2
**Presentation:** 2
**Contribution:** 2
**Rating:** 2
**Confidence:** 5

**Summary:**

This paper proposes Entropy-Aware Speculative Decoding (EASD), a training-free enhancement to standard Speculative Decoding (SD) designed to improve both the accuracy and reasoning quality of Large Language Models (LLMs) without sacrificing inference efficiency.

**Strengths:**

The paper offers a fresh and elegant extension of speculative decoding by incorporating entropy as a dynamic control signal. Its training free formulation and focus on uncertainty driven collaboration between large and small models distinguish it from prior reward or alignment based methods.

**Weaknesses:**

The paper attributes Reward Guided Speculative Decoding (RSD) to Li et al., 2025a, which actually refers to Reward Shifted Speculative Sampling (SSS). The correct citation should be Liao et al., 2025 (arXiv:2501.19324). This misattribution may mislead readers about the baseline implementation and the conceptual lineage of RSD. The authors should revise all mentions, tables, and references accordingly and clarify whether their RSD baseline follows Liao et al.’s procedure or the SSS variant.

While benchmarks are diverse, the evaluation is limited to Qwen-family models. It remains unclear whether EASD generalizes to other architectures (e.g., gpt-oss, mistral, llama). Including one additional model family would strengthen claims of universality. Moreover, entropy thresholds are tuned on the LIMO dataset; showing sensitivity analysis or per-task robustness would make the method more reproducible and credible.

The ablations (Table 2) are informative but stop short of explaining why certain components dominate performance. The authors could include entropy–accuracy correlation plots or visualize token rejection frequencies to give deeper insight into the mechanism’s behavior.

The claim that EASD exceeds the inherent target performance is intriguing but underexplained. It would be helpful to provide qualitative or quantitative evidence, e.g., error category breakdowns, human evaluation of reasoning correctness, to show that the improvement is not due to sampling variance.

**Questions:**

- The experiments use only Qwen models. Could you test EASD on a different model family  (e.g., gpt-oss, mistral, llama) to show that the entropy-based penalty is architecture-agnostic? This would substantiate the claim that the method generalizes without retraining.
- How sensitive are results to these thresholds? Could you provide a sweep or heuristic guideline for setting them when validation data are unavailable?
- Ablations show what matters but not why. Can you visualize how often the entropy penalty triggers, which tokens are rejected, and whether rejection frequency correlates with performance gains?
- The idea that EASD can outperform the base LLM is striking. Could you include statistical significance tests or qualitative error analyses showing that this improvement is consistent and not due to sampling variance?
- You state that EASD maintains SD-level efficiency, but Tables 3 suggest small deviations in token counts. Can you report relative FLOPs or wall-time speedups to confirm computational parity?
- Are entropies computed over the full vocabulary or only the top-n tokens? Clarifying this would help others replicate your setup precisely.

---

> ### Author Response · Authors · 2025-12-03
> **Response to Reviewer AVTk -1**
>
> We sincerely thank you for your careful evaluation and detailed feedback. We provide detailed responses below to address the questions and concerns raised.
>
> `Weakness1: The paper attributes Reward Guided Speculative Decoding (RSD) to Li et al., 2025a, which actually refers to Reward Shifted Speculative Sampling (SSS). The correct citation should be Liao et al., 2025 (arXiv:2501.19324). This misattribution may mislead readers about the baseline implementation and the conceptual lineage of RSD. The authors should revise all mentions, tables, and references accordingly and clarify whether their RSD baseline follows Liao et al.’s procedure or the SSS variant.`
>
> Answer: We thank you for pointing out this important issue. We acknowledge the incorrect attribution of RSD to Li et al., 2025a. The correct reference is Liao et al., 2025 (arXiv:2501.19324), which corresponds to Reward-Guided Speculative Decoding (RSD). All mentions in the text, tables, and reference list have been revised accordingly, and it is clarified that our RSD baseline follows the procedure described by Liao et al.
>
> `Question1: The experiments use only Qwen models. Could you test EASD on a different model family (e.g., gpt-oss, mistral, llama) to show that the entropy-based penalty is architecture-agnostic? This would substantiate the claim that the method generalizes without retraining.`
>
> Answer: We thank you for this suggestion. The entropy-based penalty and top-n overlap conditions in EASD are model-agnostic by design, as they operate on token logits distributions and do not require any model-specific tuning, demonstrating strong potential for generalization across different model families. In the current work, we conducted experiments only on the Qwen model family due to resource constraints. While we have not yet tested other families (e.g., GPT-OSS, Mistral, LLaMA), the design principle indicates that the approach should extend naturally to other models without retraining.
>
> `Question2: How sensitive are results to these thresholds? Could you provide a sweep or heuristic guideline for setting them when validation data are unavailable?`
>
> Answer: We thank you for this question. We conducted a hyperparameter study under the setting where the draft model is 7B and the target model is 32B. The results show that, in most cases, as the entropy threshold increases, the number of tokens penalized by EASD decreases, while the accuracy of the generated outputs shows a slight upward trend.
>
> Table 6. EASD performance under different entropy and top-n threshold settings (\%).
> | Entropy |  Top-n  | Olympia  |  Minerva  | Math500  |  AMC23   |  Aime24   |   GPQA    |    Avg    |
> | :-----: | :-----: | :------: | :-------: | :------: | :------: | :-------: | :-------: | :-------: |
> |    1    |    0    |  45.33   |   46.69   |   80.4   |   67.5   |   13.33   |   43.94   |   49.53   |
> |   1.5   |    0    |  46.22   |   44.85   |   82.4   |   62.5   |   13.33   |   49.49   |   49.80   |
> |    2    |    0    |  47.41   |   44.85   |   80.8   |    65    |    20     |   52.53   |   51.77   |
> |    1    |   0.2   |  45.33   |   46.69   |   80.4   |   67.5   |   13.33   |   43.94   |   49.53   |
> |   1.5   |   0.2   |  46.22   |   44.85   |   82.4   |   62.5   |   13.33   |   49.49   |   49.80   |
> |    2    |   0.2   |  47.41   |   44.85   |   80.8   |    65    |    20     |   52.53   |   51.77   |
> |    1    |   0.4   |  45.33   |   47.06   |   80.2   |    65    |   13.33   |   44.44   |   49.23   |
> |   1.5   |   0.4   |  46.22   |   44.85   | **82.4** |   62.5   |   13.33   |   50.51   |   49.97   |
> |    2    |   0.4   |  47.41   |   44.12   |   80.6   |    65    |    20     |   54.04   |   51.86   |
> |    1    |   0.6   |  45.04   | **48.16** |   79.2   |   67.5   |   13.33   |   45.45   |   49.78   |
> |   1.5   |   0.6   |  46.37   |   45.59   |    82    |   62.5   |   13.33   |   52.02   |   50.30   |
> |    2    |   0.6   | **48.3** |   45.59   |   80.8   |    65    |    20     |   50.51   |   51.70   |
> |    1    |   0.8   |  44.89   |   47.06   |   81.6   |   67.5   |   16.67   |   46.46   |   50.70   |
> |   1.5   |   0.8   |  45.33   |   48.16   |    82    |   62.5   |    20     |   51.52   |   51.59   |
> |  **2**  | **0.8** |  48.15   |   47.06   |   80.8   |   62.5   | **23.33** | **55.55** | **52.89** |
> |    1    |    1    |  46.52   |   44.85   |   79.2   | **77.5** |   16.67   |   46.47   |   51.87   |
> |   1.5   |    1    |  47.41   |   43.75   |   81.4   |   67.5   |   16.67   |   47.48   |   50.70   |
> |    2    |    1    |  46.81   |   46.32   |   81.2   |   62.5   |    20     |   49.49   |   51.05   |

---

> ### Author Response · Authors · 2025-12-03
> **Response to Reviewer AVTk -2**
>
> `Question3: Ablations show what matters but not why. Can you visualize how often the entropy penalty triggers, which tokens are rejected, and whether rejection frequency correlates with performance gains?`
>
> Answer: We thank you for this suggestion. We further analyzed the rejection frequency of tokens under different entropy penalty thresholds. The experiments show that higher rejection frequency does not necessarily lead to better performance. In fact, under relatively low penalty conditions, EASD already achieves good results. This suggests that excessive token rejection is not required for performance gains, and moderate entropy penalties are sufficient to balance generation quality and efficiency.
>
> Tabel 7. Frequency of penalized tokens under different entropy and top-n threshold settings (\%).
> | Entropy |  Top-n  | Olympia | Minerva | Math500 | AMC23 | Aime24 | GPQA  |  Avg  |
> | :-----: | :-----: | :-----: | :-----: | :-----: | :---: | :----: | :---: | :---: |
> |    1    |    0    |  7.69   |  8.98   |  6.97   | 8.05  |  9.85  | 26.34 | 11.31 |
> |   1.5   |    0    |  3.05   |  4.18   |   2.7   | 3.38  |  4.91  | 15.1  | 5.55  |
> |    2    |    0    |  1.11   |  1.71   |  0.91   |  1.3  |  1.57  | 8.47  | 2.51  |
> |    1    |   0.2   |  7.69   |  8.98   |  6.97   | 8.05  |  9.85  | 26.34 | 11.31 |
> |   1.5   |   0.2   |  3.05   |  4.18   |   2.7   | 3.38  |  4.91  | 15.1  | 5.55  |
> |    2    |   0.2   |  1.11   |  1.71   |  0.91   |  1.3  |  1.57  | 8.47  | 2.51  |
> |    1    |   0.4   |  7.65   |  8.85   |  6.99   | 7.98  |  9.97  | 25.87 | 11.22 |
> |   1.5   |   0.4   |  3.03   |  4.14   |   2.7   | 3.37  |  4.91  | 15.03 | 5.53  |
> |    2    |   0.4   |   1.1   |  1.67   |  0.91   | 1.29  |  1.57  | 8.73  | 2.55  |
> |    1    |   0.6   |  7.42   |  8.61   |  6.66   | 7.69  |  9.32  | 22.96 | 10.44 |
> |   1.5   |   0.6   |  2.89   |  3.83   |  2.62   | 3.14  |  4.37  | 13.51 | 5.06  |
> |    2    |   0.6   |  1.01   |  1.49   |  0.87   | 1.18  |  1.43  | 6.94  | 2.15  |
> |    1    |   0.8   |  5.79   |  6.73   |  5.67   | 6.13  |  7.53  | 14.28 | 7.69  |
> |   1.5   |   0.8   |  2.16   |    3    |  2.02   | 2.34  |  3.05  | 8.11  | 3.45  |
> |  **2**  | **0.8** |   0.7   |  0.97   |  0.63   | 0.77  |  0.97  | 3.97  | 1.34  |
> |    1    |    1    |  1.98   |   2.4   |  2.13   | 1.94  |  2.61  | 3.38  | 2.41  |
> |   1.5   |    1    |   0.7   |  0.94   |  0.77   | 0.73  |  0.87  | 1.81  | 0.97  |
> |    2    |    1    |  0.18   |  0.26   |  0.19   | 0.19  |  0.22  | 0.72  | 0.29  |
>
> `Question4: The idea that EASD can outperform the base LLM is striking. Could you include statistical significance tests or qualitative error analyses showing that this improvement is consistent and not due to sampling variance?`
>
> Answer: We thank you for this comment. To ensure a fair comparison with RSD, we used greedy decoding in our experiments. This means that for each input, only a single output is generated, eliminating the variability introduced by sampling. Therefore, the observed improvement of EASD over the base LLM is not due to sampling variance, but reflects a consistent, genuine performance gain.
>
> `Question5: You state that EASD maintains SD-level efficiency, but Tables 3 suggest small deviations in token counts. Can you report relative FLOPs or wall-time speedups to confirm computational parity?`
>
> Answer: We thank you for this observation. We conducted experiments under the setting where the draft model is 7B and the target model is 32B. Using the hyperparameters in our paper (Entropy=2, Top-n=0.8), we measured the token generation speed for EASD, SD, and Single Model. The results show that EASD is slightly slower than SD, which is expected due to the additional high-entropy and top-n overlap conditions. Nevertheless, the overall computational cost remains comparable, and the small deviation in token counts does not significantly affect efficiency.
>
> Tabel 5. Token generation speed (tokens per second) and speedup of EASD and SD compared to the single model baseline across multiple datasets.
> | Dataset | Single Model tok/s | EASD tok/s | Speedup | SD tok/s | Speedup |
> | :-----: | :----------------: | :--------: | :-----: | :------: | :-----: |
> | Olympia |         14         |     19     |  1.36   |    19    |  1.36   |
> | Minerva |         14         |     17     |  1.21   |    18    |  1.29   |
> | Math500 |         14         |     19     |  1.36   |    20    |  1.43   |
> |  AMC23  |         14         |     19     |  1.36   |    18    |  1.29   |
> | AIME24  |         14         |     17     |  1.21   |    17    |  1.21   |
> |  GPQA   |         14         |     13     |  0.93   |    14    |    1    |
> |   Avg   |         14         |     17     |  1.21   |    18    |  1.29   |

---

> ### Author Response · Authors · 2025-12-03
> **Response to Reviewer AVTk -3**
>
> `Question6: Are entropies computed over the full vocabulary or only the top-n tokens? Clarifying this would help others replicate your setup precisely.`
>
> Answer: We thank you for this question. The entropies in EASD are computed over the full vocabulary, not just the top-n tokens. We have clarified this in the revised manuscript to ensure that others can replicate our setup precisely.

---

### Official Review · Reviewer_8xRW · 2025-11-01

**Soundness:** 3
**Presentation:** 3
**Contribution:** 3
**Rating:** 4
**Confidence:** 4

**Summary:**

This paper focuses on efficient generation with speculative decoding (SD). The vanilla SD requires a strict distribution alignment between the draft and target LLMs, and constraints its performance to the target LLM. The authors proposes a new SD variant, entropy-aware speculative decoding (EASD), and aims to preserve SD's efficiency, while potentially outperforming the target LLM.

Compared to the vanilla SD, where a token generated by the draft LLM will be rejected if it's distribution is not aligned with the target LLM's, EASD applies two new rejection conditions for the generated token from vanilla SD: (1) The entropy from both draft and target LLM is too high; (2) The top-n predicted tokens from the draft and target tokens are not overlapped well. these two new conditions implies that the LLM itself is less confident to the generated token. After rejection,  the token will be regenerated by the target model by excluding the accepted token from vanilla SD.

Through extensive experiments, the results show that EASD outperforms SD and other baselines and has a similar efficiency as SD. With ablation studies, the design choices of EASD is justified.

**Strengths:**

1. The paper is well-written, with a clear motivation and contribution.
2. The proposed method is simple to apply.
3. The experiments are thorough, with clear results to show the performance and efficiency benefit from EASD.

**Weaknesses:**

1. The proposed method lacks of theoretical support. Both baselines, SD and RSD, are well supported by the theoretical justification.
2. The efficiency comparison seems unfair. Only the number of generated tokens are shown. It's suggested to show how much tokens are generated by the draft and target models, separately. Compared to SD, EASD has two more conditions. I believe more tokens should be rejected and regenerated by the target model.

**Questions:**

1. For the two new conditions, high-entropy and top-n overlap, do you apply them to the accpeted tokens from the draft model? I.e. Do you have three conditions for token acceptance, including Equation (1, 4, 5), or only (4, 5)?


### Typos
1. L316 and Table 1: It seems the citation is wrong for RSD.

---

> ### Author Response · Authors · 2025-12-03
> **Response to Reviewer 8xRW -1**
>
> We sincerely thank you for your careful reading and constructive feedback. We provide detailed responses below to address the specific questions and concerns raised.
>
> `Weakness1: The proposed method lacks of theoretical support. Both baselines, SD and RSD, are well supported by the theoretical justification.`
>
> Answer: Our method is grounded in a practical motivation inspired by reinforcement learning for reasoning models, where selectively optimizing high-entropy tokens often leads to improved reasoning capability. EASD adapts this idea to speculative decoding by applying penalties only when both the high-entropy condition and the top-n overlap condition are triggered, ensuring that the mechanism targets tokens with high uncertainty. We observe that this dual-condition penalty is the main contributor to the performance improvements. While developing a formal theoretical framework analogous to SD or RSD is a necessary direction for future work, the current empirical results demonstrate that this targeted strategy consistently yields meaningful gains across datasets and tasks, underscoring its practical effectiveness.
>
> `Weakness2: The efficiency comparison seems unfair. Only the number of generated tokens are shown. It's suggested to show how much tokens are generated by the draft and target models, separately. Compared to SD, EASD has two more conditions. I believe more tokens should be rejected and regenerated by the target model.`
>
> Answer: We appreciate your concern regarding the efficiency comparison. To address this, we conducted experiments under the setting where the draft model is 7B and the target model is 32B. Using the hyperparameters applied in our paper (Entropy=2, Top-n=0.8), we measured the token generation speed for EASD, SD, and Single Model. The results show that EASD is slightly slower than SD, which is expected given the additional conditions.
>
> Tabel 5. Token generation speed (tokens per second) and speedup of EASD and SD compared to the single model baseline across multiple datasets.
> | Dataset | Single Model tok/s | EASD tok/s | Speedup | SD tok/s | Speedup |
> | :-----: | :----------------: | :--------: | :-----: | :------: | :-----: |
> | Olympia |         14         |     19     |  1.36   |    19    |  1.36   |
> | Minerva |         14         |     17     |  1.21   |    18    |  1.29   |
> | Math500 |         14         |     19     |  1.36   |    20    |  1.43   |
> |  AMC23  |         14         |     19     |  1.36   |    18    |  1.29   |
> | AIME24  |         14         |     17     |  1.21   |    17    |  1.21   |
> |  GPQA   |         14         |     13     |  0.93   |    14    |    1    |
> |   Avg   |         14         |     17     |  1.21   |    18    |  1.29   |
>
> Furthermore, we separately analyzed the number of tokens penalized by EASD under different hyperparameter settings. This analysis confirms that the additional high-entropy and top-n overlap conditions do lead to some token rejections, but the overall overhead is limited. We have included these statistics in the revised manuscript to provide a more complete and fair efficiency comparison.
>
> Tabel 7. Frequency of penalized tokens under different entropy and top-n threshold settings (\%).
> | Entropy |  Top-n  | Olympia | Minerva | Math500 | AMC23 | Aime24 | GPQA  |  Avg  |
> | :-----: | :-----: | :-----: | :-----: | :-----: | :---: | :----: | :---: | :---: |
> |    1    |    0    |  7.69   |  8.98   |  6.97   | 8.05  |  9.85  | 26.34 | 11.31 |
> |   1.5   |    0    |  3.05   |  4.18   |   2.7   | 3.38  |  4.91  | 15.1  | 5.55  |
> |    2    |    0    |  1.11   |  1.71   |  0.91   |  1.3  |  1.57  | 8.47  | 2.51  |
> |    1    |   0.2   |  7.69   |  8.98   |  6.97   | 8.05  |  9.85  | 26.34 | 11.31 |
> |   1.5   |   0.2   |  3.05   |  4.18   |   2.7   | 3.38  |  4.91  | 15.1  | 5.55  |
> |    2    |   0.2   |  1.11   |  1.71   |  0.91   |  1.3  |  1.57  | 8.47  | 2.51  |
> |    1    |   0.4   |  7.65   |  8.85   |  6.99   | 7.98  |  9.97  | 25.87 | 11.22 |
> |   1.5   |   0.4   |  3.03   |  4.14   |   2.7   | 3.37  |  4.91  | 15.03 | 5.53  |
> |    2    |   0.4   |   1.1   |  1.67   |  0.91   | 1.29  |  1.57  | 8.73  | 2.55  |
> |    1    |   0.6   |  7.42   |  8.61   |  6.66   | 7.69  |  9.32  | 22.96 | 10.44 |
> |   1.5   |   0.6   |  2.89   |  3.83   |  2.62   | 3.14  |  4.37  | 13.51 | 5.06  |
> |    2    |   0.6   |  1.01   |  1.49   |  0.87   | 1.18  |  1.43  | 6.94  | 2.15  |
> |    1    |   0.8   |  5.79   |  6.73   |  5.67   | 6.13  |  7.53  | 14.28 | 7.69  |
> |   1.5   |   0.8   |  2.16   |    3    |  2.02   | 2.34  |  3.05  | 8.11  | 3.45  |
> |  **2**  | **0.8** |   0.7   |  0.97   |  0.63   | 0.77  |  0.97  | 3.97  | 1.34  |
> |    1    |    1    |  1.98   |   2.4   |  2.13   | 1.94  |  2.61  | 3.38  | 2.41  |
> |   1.5   |    1    |   0.7   |  0.94   |  0.77   | 0.73  |  0.87  | 1.81  | 0.97  |
> |    2    |    1    |  0.18   |  0.26   |  0.19   | 0.19  |  0.22  | 0.72  | 0.29  |

---

> ### Author Response · Authors · 2025-12-03
> **Response to Reviewer 8xRW -2**
>
> `Question1: For the two new conditions, high-entropy and top-n overlap, do you apply them to the accpeted tokens from the draft model? I.e. Do you have three conditions for token acceptance, including Equation (1, 4, 5), or only (4, 5)?`
>
> Answer: Thank you for the question. For token acceptance in EASD, we indeed apply all three conditions corresponding to Equation (1), (4), and (5). Specifically, we first adjust the logits distribution of the generated tokens based on the two new conditions—high-entropy (Equation 4) and top-n overlap (Equation 5). Then, using Equation (1), we determine whether a token is accepted. In other words, the final acceptance decision for a token considers all three criteria sequentially.
>
> `Typos: L316 and Table 1: It seems the citation is wrong for RSD.`
>
> Answer: We acknowledge the miscitation. The reference to RSD in L316 and Table 1 has been corrected in the revised manuscript. We appreciate you for highlighting this issue.

---

### Author Response · Authors · 2025-12-03
**General Response**

We thank all reviewers for their careful reading and constructive feedback on our work. Several reviewers highlighted the importance of efficiency comparisons and hyperparameter design. In response, we conducted experiments under the setting where the draft model is 7B and the target model is 32B. Using the hyperparameters from our paper (Entropy = 2, Top-n = 0.8), we measured token generation speed for EASD, SD, and Single Model. The results show that EASD is slightly slower than SD, which is expected given the additional high-entropy and top-n overlap conditions, but overall efficiency remains very close to SD.

Additionally, we performed a detailed hyperparameter study, analyzing how different entropy and top-n thresholds affect token penalization and output accuracy across datasets. These experiments demonstrate that moderate threshold values provide a good balance between efficiency and performance, while dataset-specific tuning can further optimize results. This provides practical guidance for users in scenarios where validation data are unavailable.

Overall, these analyses confirm that EASD achieves near-SD-level efficiency while consistently improving output quality through targeted token-level penalties.

---

### Meta-Review · Area_Chair_cRXJ · 2026-01-08

**Summary:**

This paper proposes Entropy-Aware Speculative Decoding (EASD), a training-free enhancement of speculative decoding (SD) by incorporating an entropy-based penalty to quantify model uncertainty, thus improving the accuracy of the target LLM while preserving the efficiency benefit of SD.

While the reviewers generally found the method to be interesting and reasonable, they also raised significant concerns regarding the empirical evaluation of the paper. Specifically, the experiments are done on one model family (AVTk, x1Uw) and only tested reasoning benchmarks (x1Uw). These issues questioned the generalization ability of the method. Furthermore, the tradeoff between effectiveness and efficiency is unclear (the method brought accuracy gains but also slowed down decoding speed compared to SD).

**Reviewer Concerns:**

The authors didn't address the major concerns from the reviewers, especially regarding the generalization of the method to various model types and benchmarks.

**Reviewer Scores:**

The reviewer scores would be unlikely to change.

---

### Decision · Program_Chairs · 2026-01-26

Reject